# Impact of Diet on Plasma Lipids in Individuals with Heterozygous Familial Hypercholesterolemia: A Systematic Review of Randomized Controlled Nutritional Studies

**DOI:** 10.3390/nu13010235

**Published:** 2021-01-15

**Authors:** Gabrielle Roy, Anykim Boucher, Patrick Couture, Jean-Philippe Drouin-Chartier

**Affiliations:** 1Centre Nutrition, Santé et Société (NUTRISS), Institut Sur la Nutrition et les Aliments Fonctionnels (INAF), Université Laval, Québec, QC G1V 0A6, Canada; gabrielle.roy.12@ulaval.ca (G.R.); anykim.boucher.1@ulaval.ca (A.B.); patrick.couture@fmed.ulaval.ca (P.C.); 2Centre de Recherche du CHU de Québec-Université Laval, Québec, QC G1V 4G2, Canada; 3Faculté de Pharmacie, Université Laval, Québec, QC G1V 0A6, Canada

**Keywords:** systematic review, familial hypercholesterolemia, diet, intervention, dietary supplement, food, nutrients

## Abstract

Background: Conclusive data on the effectiveness of dietary interventions in heterozygous familial hypercholesterolemia (HeFH) management are unavailable. Whether this is due to a true lack of effects or biases in intervention designs remains unsettled. We systematically assessed the impact on LDL-C of published dietary randomized controlled trials (RCTs) conducted among individuals with HeFH in relation to their design and risk of bias. Methods: We systematically searched PubMed, Web of Science, and Embase in November 2020 to identify RCTs that assessed the impact of: (1) food-based interventions; (2) dietary counseling interventions; or (3) dietary supplements on LDL-C in individuals with HeFH. We evaluated the risk of bias of each study using the Cochrane Risk of Bias 2 method. Results: A total of 19 RCTs comprising 837 individuals with HeFH were included. Of those, five were food-based interventions, three were dietary counseling interventions and 12 were dietary supplement-based interventions (omega-3, *n* = 3; phytosterols, *n* = 7; guar gum, *n* = 1; policosanol, *n* = 1). One study qualified both as a food-based intervention and as a dietary supplement intervention due to its factorial design. A significant reduction in LDL-C levels was reported in 10 RCTs, including eight dietary supplement interventions (phytosterols, *n* = 6, omega-3, *n* = 1; guar gum, *n* = 1), one food-based intervention and one dietary counseling intervention. A total of 13 studies were judged to have some methodological biases in a way that substantially lowers confidence in the results. Studies at low risk of biases were more likely to report significant reductions in LDL-C concentrations, compared with studies at risk of bias (chi-square statistic: 5.49; *p* = 0.02). Conclusion: This systemic review shows that the apparent lack of effectiveness of diet manipulation in modulating plasma levels of LDL-C among individuals with HeFH is likely due to biases in study designs, rather than a true lack of effects. The likelihood of reporting significant reductions in LDL-C was associated with the concurrent risk of bias.

## 1. Introduction

Heterozygous familial hypercholesterolemia (HeFH) is an autosomal dominant disease caused by genetic mutations in the LDL receptor (LDLR), its ligand apolipoprotein B (Apo B), or proprotein convertase subtilisin/kexin type 9 (PCSK9) [1]. These mutations disrupt the normal clearance of LDLs from the plasma, causing a marked hypercholesterolemia across the lifespan and accelerated atherosclerosis [2]. If untreated, individuals with HeFH face a 10- to 20-fold increased risk in coronary heart disease (CHD) compared with unaffected individuals [3]. With an estimated prevalence of 1 in 310 individuals [4], HeFH is the most common genetic disorder causing premature coronary events and deaths worldwide [1].

In HeFH management, the primary objective is to decrease and maintain low LDL-cholesterol (C) concentrations in order to reduce atherosclerosis progression and long-term risk of CHD [1]. Given the genetic nature of this disease, diet has traditionally been considered as a secondary therapy, with limited effectiveness relative to medication [1,5]. Even though the literature on diet in HeFH management remains limited, previous systematic reviews and meta-analyses of dietary interventions among individuals with HeFH suggested that no conclusions can be made about the effectiveness of diet in HeFH [6,7]. Still, none of the previous reviews thoroughly assessed the impact on plasma lipids of dietary interventions conducted in HeFH, in line with the intervention design per se [6,7]. This element is critical to investigate, as food-based interventions, dietary counseling studies and dietary supplement trials are exposed to specific biases (e.g., food substitution effect [8], adherence to dietary advice [9] or compliance with supplementation protocol [10]) which are likely to have different, yet direct impacts on their effectiveness in modulating plasma lipids [11]. These potential issues need to be taken into consideration when interpreting results in order to adequately appreciate the role of diet in HeFH management, and orient future research in the field. In summary, whether the apparent lack of effectiveness of dietary interventions in modulating plasma lipids in HeFH is due to a true lack of effects or to interventions’ design and associated biases remains unsettled [6,7].

Therefore, the objective of the current study was to systematically review the impact on plasma lipids of previously published randomized controlled nutritional studies (RCTs) conducted among individuals with HeFH according to their design and inherent biases. Specifically, we systematically assessed the impact of published food-based interventions, dietary counseling interventions and dietary supplement trials on LDL-C and other plasma lipids in children or adults with HeFH in relation to their risk of bias.

## 2. Materials and Methods

The present review is reported according to the Preferred Reporting Items for Systematic Reviews and Meta-Analyses statement (PRISMA) [12]. We registered the protocol on the international prospective register of systematic reviews (PROSPERO, # CRD42020189706).

### 2.1. Study Eligibility Criteria

Studies were included if: (1) they were RCTs conducted among individuals with HeFH; (2) they assessed the impact of a dietary intervention (i.e., food-based intervention, nutritional counseling intervention or dietary supplement trial) on plasma lipids, including LDL-C; (3) they provided head-to-head post-intervention comparison between the intervention and the control arms for LDL-C concentrations; or (4) they were published in English. The rationale for focusing study selection on LDL-C was based on the fact that this lipid fraction is the main treatment target in HeFH [1].

### 2.2. Search Strategy

A systematic search was conducted in PubMed, Embase and Web of Science on 25 November 2020. Details of the searches in each database are presented in Appendix A. No restriction was applied on publication date or publication status. Two authors (A.B. and G.R.) independently screened the literature to identify relevant articles. References cited in all included studies were additionally screened to identify potentially relevant studies not captured by the database searches. Disagreement and discordance between the two authors was resolved by discussion. When no agreement could be obtained, a third author (J.-P.D.-C) was consulted.

### 2.3. Data Extraction

Data extraction was performed by a single author (G.R.) and audited by a second author (A.B.). For each included study, extracted data included the author’s first name, year of publication, the country where the study was conducted, the number, sex and age of participants, study design (including whether it was designed as a crossover or parallel intervention, and information on blinding of study participants and investigators, if applicable), intervention characteristics (e.g., duration, characteristics of the intervention and control arms, information on concomitant medication) and post-intervention plasma lipid concentrations.

### 2.4. Assessment of the Risk of Bias within Studies

Two authors (A.B and G.R.) independently assessed the risk of bias of included studies using the second version of the Cochrane Risk-of-Bias instrument (RoB 2) [13]. Risk of bias was assessed in terms of the following domains: (1) randomization process; (2) deviations from intended interventions; (3) missing outcomes data; (4) measurement of the outcome; (5) selective outcome reporting; and (6) overall bias. Each domain was judged at “low risk”, “some concerns” or “high risk” of bias. According to RoB 2, a judgment of “high risk” of bias for any individual domain leads to an overall high risk of bias. The same rationale applies to domains judged at “some concerns” [13]. However, if methodological concerns are raised in multiple domains in a way that substantially lowers confidence in the result, the overall risk of bias is to be judged “high”. Any discrepancies in judgment were resolved by discussion and consensus between the two authors, or by consulting a third author (J.-P.D.-C) when necessary.

### 2.5. Data Synthesis 

Studies were primarily grouped and analyzed according to their design (i.e., food-based intervention, dietary counseling intervention or dietary supplement intervention). We emphasized the review on the impact of the identified dietary interventions on concentrations of LDL-C, Apo B and Total-C. The impact of the interventions on other plasma lipids (e.g., TGs, HDL-C, Lp(a)) were also synthesized depending on data availability. Lastly, we compared the likelihood of the identified studies of reporting significant reductions in LDL-C, according to the risk of bias of each study using a chi-square test.

## 3. Results

### 3.1. Study Selection

As shown in Figure 1, 1911 research articles were identified through database searching (PubMed, *n* = 553; Embase, *n* = 491; Web of Science, *n* = 867). After removing duplicates, 599 unique articles were screened. Following title and abstract examination, the full text of 78 articles was reviewed. One article [14] was identified by screening the list of references of a relevant study. A total of 19 unique articles met the inclusion criteria, and were thus included in the current systematic review [14,15,16,17,18,19,20,21,22,23,24,25,26,27,28,29,30,31,32].

### 3.2. Characteristics of Included Studies

Characteristics of the 19 included studies are presented in Table 1. There were five food-based interventions [15,16,17,18,19], three dietary counseling interventions [20,21,22] and 12 dietary supplement interventions (omega-3, *n* = 3; phytosterols, *n* = 7; guar gum, *n* = 1; policosanol, *n* = 1) [14,19,23,24,25,26,27,28,29,30,31,32]. Notably, the study by Fuentes et al. [19] qualified both as a food-based controlled intervention and as a phytosterol supplementation intervention because of its factorial design. A total of 16 studies had a crossover design [14,15,16,17,18,19,20,23,24,25,26,27,28,30,31,32], and three had a parallel design [21,22,29]. Thirteen studies included exclusively adult participants [14,15,16,18,19,20,21,23,24,26,29,31,32], while six studies were conducted among children [17,22,25,27,28,30]. The duration of the interventions ranged from three weeks [16] to 52 weeks [21]. In 10 studies, participants were using lipid-lowering medication during the intervention [14,18,19,20,21,23,24,26,29,31], whereas in the nine others [15,16,17,22,25,27,28,30,32], the interventions were conducted without concomitant lipid-lowering medication.

### 3.3. Food-Based Interventions (n = 5)

Gustafsson et al. [15] compared the effect of a fully controlled diet with a polyunsaturated-to-saturated fat ratio of 2.0 (treatment arm) to a fully controlled diet with a polyunsaturated-to-saturated fat ratio of 1.3 (control arm) for six weeks using a randomized crossover design. Six adults with HeFH not treated with lipid-lowering medication were enrolled in the study. No significant difference between the two diets was observed in post-diet concentrations of LDL-C or other plasma lipids (Table 1).

Friday et al. [16] conducted a crossover study where five adults with HeFH not treated with lipid-lowering medication received a fully controlled diet enriched in (1) safflower oil, (2) salmon oil diet or (3) butter (control arm) in a random order. Each diet lasted three weeks. Compared with the butter diet, the safflower and salmon oil diets significantly decreased plasma concentrations of LDL-C (–28.4% and –31.5%, respectively; *p* < 0.001), Total-C (–25.3% and –34.7%, respectively; *p* < 0.001), and Apo B (–26.0% and –28.4%, respectively; *p* < 0.01). The salmon diet also induced significant decreases in plasma concentrations of HDL-C (–12.6%; *p* = 0.003) and triglycerides (TGs) (–46.3%; *p* = 0.03) compared with the butter diet (Appendix A). The safflower diet had no significant impact on HDL-C and TG concentrations.

Using a crossover design, Laurin et al. [17] compared the effects of a fully controlled diet where 35% of protein intake was provided from either a soy beverage (treatment arm) or cow’s milk (control arm) for four weeks in 10 children with HeFH not using concomitant lipid-lowering medication. No significant difference was found in plasma levels of LDL-C and other Apo B-containing lipoproteins. However, compared with the cow’s milk diet, the diet supplemented with the soy beverage induced a significant increase of 4.3% (*p* < 0.04) in HDL-C. Two subjects out of the 10 initially enrolled were removed from the analyses by the investigators due to low compliance and elevated serum lipid levels. This induced a high risk of bias in the domain on missing outcome data.

Wolfe et al. [18] conducted a crossover study where five adults with HeFH were assigned to a fully controlled high-protein diet (27% of energy provided as proteins and 48% as carbohydrates; treatment arm) and a fully controlled low-protein diet (10% of energy provided as protein and 65% as carbohydrates; control arm) in a random order for four to five weeks while being treated with cholestyramine. No difference was observed in post-diet concentrations of plasma lipids. 

Finally, Fuentes et al. [19] compared the effect of a fully controlled diet providing 150 mg of dietary cholesterol/day (treatment arm) to a fully controlled diet providing 280–300 mg of dietary cholesterol/day (control arm) in 30 adults with HeFH for four weeks using a crossover, factorial design. Participants were concomitantly treated with statins. No difference in plasma lipid concentrations was observed between the low- and high-cholesterol diets at the end of the intervention.

As shown in Table 2, methodological concerns were identified in all of the above studies, mainly due to a lack of information on the randomization process and on the blinding of the participants and/or study staff. Some concerns were identified in the overall risk of bias in four studies [15,16,18,19], whereas the one by Laurin et al. [17] was judged to be at high risk of bias due to the selective removal of participants prior to the analyses.

### 3.4. Dietary Counseling Interventions (n = 3)

Chisholm et al. [20] conducted a crossover study where 19 adults with HeFH treated with simvastatin received dietary counseling to consume a low-fat diet (treatment arm) and a high-fat diet (control arm) in a random order. Each arm lasted eight weeks. Participants received nutritional counseling every 15 to 30 days by an experienced dietician. In the high-fat diet arm of the study, participants did not reach the targeted level of daily energy consumed from fat (33% of total energy instead of 38%). This induced a high risk of bias in the domain on deviations from the intended intervention (Table 2). A significant reduction in LDL-C (–6.4%; *p <* 0.05) and Total-C (–6.3%; *p* < 0.01) concentration was observed after the low-fat diet compared with the high-fat diet. The low-fat diet also significantly decreased plasma concentrations of HDL-C (–7.6%; *p* < 0.05) compared with the high-fat diet (Appendix A).

Broekhuizen et al. [21] conducted a parallel intervention where 340 adults with HeFH, including 136 concomitantly treated with lipid-lowering medication, were randomized to either a personalized health counseling intervention (treatment arm) or care as usual (control arm) for 52 weeks. No difference was observed in concentrations of LDL-C, Total-C and Apo B between the intervention and the control group after 52 weeks. What is noteworthy is that only 49% of the participants completed at least one of the advice modules presented on the website. This induced a high risk of bias in the domain on deviations from the intended intervention.

Helk et al. [22] conducted a parallel trial during which 34 children with HeFH (not treated with lipid-lowering medication) received nutritional counseling to consume either a diet high in unsaturated fats, low in saturated fats and enriched with soy protein (intervention arm) or a diet high in unsaturated fats and low in saturated fats (control arm) for 13 weeks. No difference was observed in post-intervention concentrations of plasma lipids. During the study, four subjects dropped out. Prior to statistical analyses, investigators removed four other participants due to compliance issues. The latter induced a high risk of bias in the domain on missing outcomes data.

As shown in Table 2, the overall risk of bias was judged to be at high risk in all of the studies on nutritional counseling interventions [20,21,22].

### 3.5. Dietary Supplement Interventions (n = 12)

Of the 12 included studies that assessed the effect of a dietary supplement on LDL-C and other plasma lipids [14,19,23,24,25,26,27,28,29,30,31,32], three investigated the effects of omega-3 supplements [14,23,24], seven pertained on phytosterol supplements [19,25,26,27,28,29,30], one assessed the impact of guar gum [31] and one evaluated the impact of policosanol (i.e., a mixture of higher primary aliphatic alcohols isolated from sugar cane wax) [32].

### 3.6. Omega-3 Supplementation (n = 3)

Balestrieri et al. [23] conducted a double-blinded crossover study where 16 adults with HeFH consumed fish oil ester capsules (6 g/day) or olive oil capsules (6 g/day) in a random order for four weeks. Participants were concomitantly using simvastatin. Omega-3 supplementation had no impact on plasma lipids compared with olive oil supplements. Two participants dropped out from the trial due to medical reasons related to the outcome (heart attack and vascular surgery), and were not included in the final analysis. This introduced some concerns for bias regarding the third domain of bias (missing outcome data). 

Chan et al. [24] conducted a crossover study where 20 adults with HeFH had to supplement their diet with omega-3 fatty acid ethyl ester supplements (4 g/day), or consume their usual diet for 16 weeks in a random order. No placebo capsules were provided to participants during the control arm, which introduced a high risk of bias in the domain of deviations from intended intervention. During the trial, patients were concomitantly treated with lipid-lowering medication (statin, or statin plus ezetimibe). Omega-3 supplementation had no impact on LDL-C and Total-C concentrations, but significantly decreased plasma levels of Apo B (–8.4%; *p* = 0.04) and TG (–19.2%; *p* = 0.01), compared with the control arm of the study.

Hande et al. [14] conducted a double-blinded crossover trial where 38 adults consumed omega-3 fatty acid supplements (4 g/day) or olive oil capsules (4 g/day) in a random order for three months while being treated with statins. Significant decreases in post-supplementation concentrations of LDL-C (–12.5%; *p* < 0.01), Total-C (–8.0%; *p* < 0.05) and TGs (–26.7%; *p* < 0.0001), compared with the olive oil supplements, were reported. 

The overall risk of bias was judged to be of some concern in the study by Balestrieri et al. [23], at high risk of bias in the one by Chan et al. [24], and at low risk of bias in the study by Hande et al. [14] (Table 2). 

### 3.7. Phytosterols Supplementation (n = 7)

Gylling et al. [25] evaluated the effect of sitostanol esters (3 g/day) dissolved in rapeseed oil margarine compared with a placebo margarine for six weeks in 14 children with HeFH in a double-blinded crossover fashion. Children were not treated with lipid-lowering medication. The sitostanol-enriched margarine induced significant decreases in plasma LDL-C (–15.0%; *p* < 0.05) and Total-C (–10.6%; *p* < 0.05) levels, compared with the placebo margarine. 

Neil et al. [26] conducted a double-blinded crossover intervention where 62 adults with HeFH concomitantly treated with statins supplemented their diet with a plant sterol-enriched fat spread, providing 2.5 g/day of sterol esters and a placebo fat spread free of sterol in a random order for eight weeks. Significant reduction in plasma concentrations of LDL-C (–6.8%; *p* = 0.001), Total-C (–5.0%; *p* = 0.001) and HDL-C (–4.2%; *p* < 0.01) were observed at the end of the phytosterol supplementation, compared with the placebo spread.

Amundsen et al. [27] enrolled 41 children with HeFH in a double-blinded crossover trial in which participants had to consume a plant sterol-enriched fat spread providing 1.6 g/day of sterol esters and a placebo fat spread free of sterol in a random order for eight weeks. Cholesterol-lowering medication was stopped eight weeks before randomization. Significant reduction in LDL-C (–10.7%; *p* = 0.03), Total-C (–8.2%; *p* = 0.007) and Apo B (–10.8%; *p* = 0.02) were observed at the end of the sterol supplementation, compared with the placebo spread period. 

In a double-blinded crossover study, de Jongh et al. [28] compared the effect of plant sterol spread, providing 2.3 g/day of sterol esters with a sterol-free placebo margarine for four weeks in 41 children with HeFH not treated with lipid-lowering medication. Significant reductions in LDL-C (–15.2%; *p* < 0.001) and Total-C (–11.2%; *p* < 0.001) were observed at the end of the sterol supplementation, compared with the placebo period. 

O’Neill et al. [29] conducted a parallel study with a factorial design where 69 adults with HeFH were randomly assigned to one of three interventions, during which they had to supplement their diet with: (1) a plant sterol-enriched fat spread providing 1.6 g/day of sterol esters and a placebo cereal bar; (2) a plant sterol-enriched fat spread providing 1.6 g/day of stanol esters and a placebo cereal bar; or (3) a plant stanol-enriched fat spread providing 1.6 g/day of stanol esters and a cereal bar providing 1.0 g/day of stanol esters. Each study arm lasted eight weeks, and was preceded by one month on a placebo margarine (control arm). No placebo bar was given in the control arm. All participants were concomitantly treated with lipid-lowering medication (statins, or statins plus bile acid sequestrants). All three diets led to a significant reduction in LDL-C concentrations compared with the placebo margarine (sterol: –4.7%, *p* = 0.003; low-stanol: –7.6%, *p* = 0.03; high-stanol: –12.5%, *p* < 0.001). Significant reductions in Total-C concentrations were also observed. The high-stanol supplementation significantly decreased concentrations of HDL-C and TGs levels, compared with the placebo margarine (Appendix A). Having the run-in period used as the control arm induces a high risk of bias in the domain related to the measurement of the outcome. The lack of a placebo cereal bar also introduced a high risk of bias in the domain related to deviations from the intended intervention. 

In a double-blinded crossover study, Jakulj et al. [30] assigned 41 children with HeFH to consume daily a low-fat yogurt enriched with 2.0 g of stanol esters and a low-fat placebo yogurt free of stanol esters in a random order for four weeks. Participants were not receiving concomitant lipid-lowering medication. Compared with the placebo arm, significant decreases in LDL-C (–9.0%; *p* < 0.001) and Total-C (–7.6%; *p* < 0.001) concentrations were observed after the stanol supplementation.

Finally, Fuentes et al. [19] compared the impact of a fully controlled diet enriched in sterol esters provided through an enriched fat spread (2.5 g/day) to a fully controlled diet free of sterol in 30 adults with HeFH as part of a larger factorial crossover study. Participants were not treated with lipid-lowering medication. After 16 weeks, no significant differences were observed in plasma lipids between the two diets. 

As shown in Table 2, methodological concerns were identified in two of the seven studies on phytosterol supplements. The study by O’Neill et al. [29] was judged to be at a high risk of bias due to potential deviations from the intended interventions and measurement of the outcome. The study by Fuentes et al. [19] was judged to be have some concerns of bias, due to a lack of information on the randomization process and on the blinding of the participants and study staff. Other studies were at low risk of bias.

### 3.8. Other Types of Dietary Supplements (n = 2)

Wirth et al. [31] conducted a crossover trial where 12 adults with HeFH were assigned to consume guar gum supplement (15.6 g/day) in combination with bezafibrate (intervention arm) and bezafibrate alone (control arm) in a random order for eight weeks. No placebo in replacement of the guar gum was provided in the control arm. Guar gum supplements induced significant decreases in LDL-C (–13.9%; *p* < 0.01) and in Total-C (–6.3%; *p* < 0.01) concentrations. The lack of a placebo induced a high risk of bias in the domain related to deviations from intended intervention. The overall risk of bias was therefore judged to be high (Table 2). 

Greyling et al. [32] allocated 16 adults with HeFH to use policosanol supplements providing 20 mg/day of this higher aliphatic primary alcohol mixture, or a placebo in a random order for 12 weeks in a double-blinded crossover fashion. Policosanol supplementation had no impact on plasma lipids. This study was judged to be at low risk of bias (Table 2). 

### 3.9. LDL-C Modification vs. Risk of Bias

Figure 2 presents results from included studies according to their respective risk of biases. As shown in panel A, studies at low risk of bias were more likely to report significant reduction in LDL-C levels, compared with those at some concern or high risk of bias (chi-square statistic = 5.49; *p =* 0.02). This pattern was also observed when studies were additionally classified according to their design (panel B).

## 4. Discussion

In this systematic review, we assessed whether the impact on plasma lipids of previously published dietary interventions conducted among individuals with HeFH was associated with study design and quality. Among the 19 unique studies we identified and analyzed, a significant reduction in LDL-C was observed in 10 of these, including eight dietary supplement interventions, one food-based intervention and one dietary counseling intervention. As much as 13 studies were judged to have some methodological biases in a way that substantially lowers confidence in the results. Overall, dietary interventions at low risk of bias were more likely to report significant reductions in LDL-C, compared with those at risk of bias. Our review suggests that the lack of conclusive data on the effectiveness of diet manipulation in decreasing LDL-C levels in HeFH is likely due to biases in previously conducted studies rather than a true lack of effects. In this context, diet should be recognized as a factor significantly affecting plasma lipids in HeFH.

Seminal studies unequivocally demonstrated that diet is a major determinant of CHD risk across different populations, including individuals with a high genetic susceptibility to CHD [33,34,35]. Moreover, multiple shorter-term RCTs demonstrated that diet also has an important impact on plasma lipids [36,37,38]. Still, in HeFH management, diet has traditionally been considered as a secondary and often inefficient therapy [5]. For instance, the 2018 Canadian Cardiovascular Society statement on FH states that “conclusive data regarding the effectiveness of dietary interventions in FH are unavailable” [1]. In that regard, our systematic review first revealed that the lack of effectiveness of dietary interventions in modulating LDL-C levels in HeFH is likely due to biases in study designs, rather than a true lack of effects. Indeed, published RCTs with a low risk of bias were more likely to report significant reductions in LDL-C. Conversely, most RCTs with some risk or high risk of bias reported no effect of diet on LDL-C. The null effects observed in these studies thus need to be interpreted with caution, as inherent methodological biases raise uncertainty on the validity of these results. 

Notably, our work also highlighted that, in adequate study settings, diet appears to impact LDL-C and other plasma lipids similarly among individuals with HeFH than among non-FH individuals – a finding that corroborates observational data on the relationship between dietary intakes and plasma lipids in HeFH [5,39,40]. Indeed, six of the seven interventions on phytosterol (sterols and/or stanols) supplementation we reviewed reported significant reductions in LDL-C among individuals with HeFH, ranging from 5 to 15% [7]. These reductions are consistent with mean effects on LDL-C reported in meta-analyses on phytosterol supplementation interventions conducted among non-FH individuals [41]. Similarly, substituting unsaturated fatty acids for saturated fatty acids in the food-based intervention conducted by Friday et al. [16] decreased LDL-C levels by about 30%. Even though this study was judged to have some concerns in two methodological domains, and that the findings may thus be considered anecdotal, the observed cholesterol-lowering effect associated with dietary fat quality manipulation was also consistent with data from studies conducted among non-FH individuals [38]. Lastly, with regard to omega-3 supplementation, no significant effect on LDL-C levels was reported in two of the three retrieved studies, but reductions in TG levels of >15% were reported in all of them. Again, these findings are consistent with data among non-FH individuals, as omega-3 fatty acids, relative to other unsaturated fatty acids, are recognized to have lower TGs levels, not LDL-C [42]. 

With regard to dietary counseling interventions, only one [20] of the three studies [20,21,22] we identified reported significant reductions in LDL-C. However, these data should not be interpreted as a lack of effects of diet on plasma lipids, since the extent to which dietary counseling interventions may influence plasma lipids is intrinsically related to participants’ adherence to the experimental dietary recommendations. For instance, in the study by Chisholm et al. [20], participants did not reach the targeted level of daily energy consumed from fat in the control arm of the study. In the study by Broekhuizen et al. [21], only 49% of the participants completed at least one of the advice modules on the website. These adherence issues reflect the complexity of underlying behavior changes in lifestyle medicine [8,43]. Therefore, the lack of effect on LDL-C reported in these studies should be interpreted in the context of a low adherence of study participants to the prescribed interventions.

Lastly, our systemic review highlighted that the impact of overall healthy dietary patterns (e.g., Mediterranean diet, DASH diet, plant-based diet, etc.) on plasma lipids in HeFH has never been investigated. Indeed, the food-based interventions we retrieved evaluated the impact of modifications in either dietary fat or protein intakes [15,16,17,18,19], but none focused on the cholesterol-lowering effects of high-quality diets. This element is crucial to underscore, as dietary patterns integrate interactive and cumulative associations of different dietary components, and are thus likely to induce greater beneficial effects not only on plasma lipids, but also on inflammation, glucose homeostasis and blood pressure, all important CHD risk factors in HeFH [44,45]. In that regard, our group recently provided the first demonstration that diet quality, assessed with the Alternative Healthy Eating Index (AHEI), is inversely associated with coronary artery calcification prevalence and severity in adults with HeFH concomitantly treated with cholesterol-lowering medication [46]. A 15% increase in AHEI score was associated with 43% lower odds of coronary artery calcification, independent of the LDLR genotype, LDL-C year-score (i.e., an individual’s lifetime exposure to LDL-C), statin use, age, sex, body mass index and hypertension. This finding is major, as the AHEI has been consistently associated with lower CHD risk in multiple populations [47], and because coronary artery calcification is one of the most discriminant risk factors of incident CHD in HeFH [48]. Most importantly, despite the fact that the observational design precludes from inferring causality, these data strongly support that diet is a key behavioral risk factor to consider in CHD prevention in HeFH. Still, as evidenced in the current review, to date, high-quality dietary interventions in HeFH are sparse, and this has serious implications in HeFH management. On the one hand, the use of cholesterol-lowering medication dramatically improves prognosis [49,50]. On the other hand, a large inter-individual variability in the response to statins is reported, with only 50% of HeFH patients achieving LDL-C targets [51]. This exposes a significant proportion of patients to a high residual CVD risk [52,53]. Furthermore, the perceived effectiveness of medication has led to a devaluation of the importance of adopting a healthy diet in HeFH [54]. Indeed, the current drug-centered approach has been documented to be a barrier to healthy eating habits in individuals with HeFH [54]. Affected individuals tend to engage in unhealthy dietary behaviors due to this misconception about diet, which is likely to negatively impact pharmacotherapy efficacy and cardiovascular health [54,55]. There is thus a true need for high quality trials on the dietary impact on cardiovascular health in HeFH, to unsettle this misconception about diet in this high-risk population. 

Results from our systematic review need to be interpreted in the context of different strengths and limitations. First, the consideration for the intervention design in the interpretation of results is a major strength, since food-based interventions, dietary counseling interventions or dietary supplement interventions are likely to yield different results given their impact on overall diet, and inherent risk of biases [8,9,10]. The use of the RoB 2 tool in the risk of bias assessment is another strength of our review. This method reflects contemporary understanding of how methodological biases may influence study results [13]. The new features incorporated in RoB 2, including an overall risk of bias domain, signaling questions that standardize the judgment process, as well as consideration for baseline imbalances in the randomization process, improve the risk of bias assessment compared with the original version [13]. On the other hand, it must be acknowledged that the limited number of included studies precludes from drawing conclusions on whether the dietary impact on LDL-C and other plasma lipids among individuals with HeFH differ according to sex, age or the concomitant use of lipid-lowering medication. Finally, we limited our review to the dietary impact on plasma lipids, with an emphasis on LDL-C levels – the primary treatment target in HeFH management. However, reviewing the impact of diet on other CHD risk factors should also be thoroughly conducted, as these also influence cardiovascular health in individuals with HeFH. 

## 5. Conclusions

In conclusion, results from our systemic review show that the apparent lack of effectiveness of diet manipulation in modulating plasma levels of LDL-C and other lipids among individuals with HeFH is likely due to biases in study designs, rather than a true lack of effects. We observed that the likelihood of reporting significant reductions in LDL-C was significantly associated with the concurrent risk of bias. The present systematic review highlights the need to conduct well-designed RCTs assessing the impact of healthy dietary patterns on cardiovascular health in individuals with HeFH.

## Figures and Tables

**Figure 1 nutrients-13-00235-f001:**
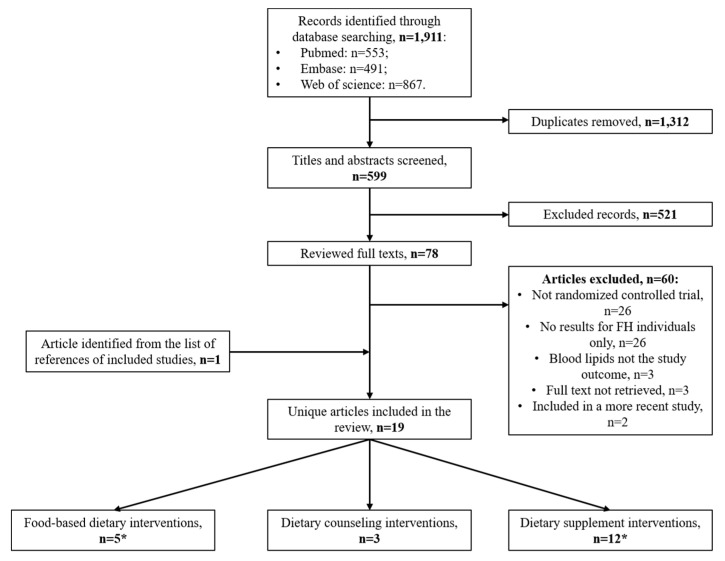
Flow chart of study selection. * The study by Fuentes et al. [19] qualified both as a food-based and a dietary supplement intervention.

**Figure 2 nutrients-13-00235-f002:**
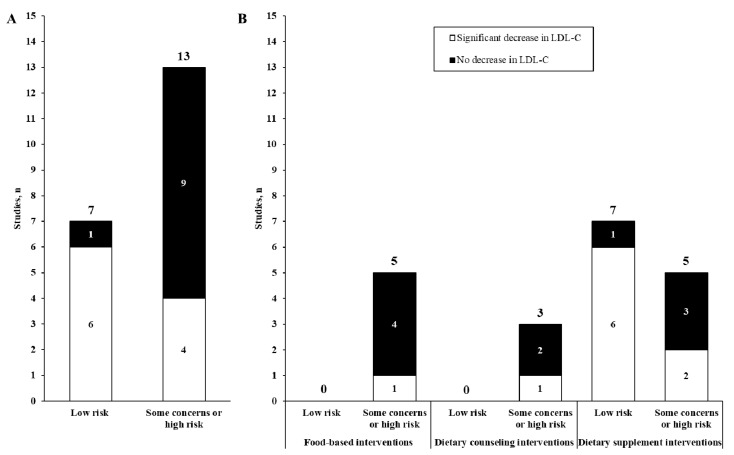
Distribution of the included studies according to their risk of bias (panel (**A**)), their design (panel (**B**)) and the reported effect on LDL-cholesterol (LDL-C) concentrations. The vertical bars represent the count of studies and the color denotes the reported effect on LDL-cholesterol (black: no decrease; white: significant decrease). In panel A, the chi-square statistic for the likelihood of reporting a significant reduction in LDL-cholesterol was 5.49 (*p* = 0.02).

**Table 1 nutrients-13-00235-t001:** Characteristics of studies included in the systematic review.

Author, Year	Country	Participants, *n*	Sex, *n*	Age (y)	Study Design	Concomitant Medication	Duration	Control Arm	Intervention Arm	Outcomes	Post-Treatment Concentration ^1^		Difference (%)	*p* Value
											Control	Intervention		
Food-based interventions														
Gustafsson et al., 1983 [15]	Sweden	6 adults	3 Females (F)/3 Males (M)	F: 50–59 M: 30–60	Crossover	None	6 weeks	Fully controlled diet with a polyunsaturated-to-saturated fat ratio of 1.3	Fully controlled diet with polyunsaturated-to-saturated fat ratio of 2.0	LDL-C	5.51	5.49	−0.4	NS
										Apo B	1.75	1.73	−1.1	NS
Friday et al., 1991 [16]	USA	5 adults	2F/3M	34 ± 6	Crossover (3 arms)	None	3 weeks	Fully controlled diet with 82.4 g of butter/2000 kcal/day	1. Fully controlled diet with 67 g of safflower oil/2000 kcal/day	LDL-C	6.72 ± 0.59	4.81 ± 0.52	−28.4	<0.001
										Total-C	9.39 ± 0.67	7.01 ± 0.54	−25.3	<0.001
										Apo B	2.08 ± 0.19	1.54 ± 0.18	−26.0	0.002
									2. Fully controlled diet with 67 g of salmon oil/2000 kcal/day	LDL-C	6.72 ± 0.59	4.60 ± 0.47	−31.5	<0.001
										Total-C	9.39 ± 0.67	6.13 ± 0.39	−34.7	<0.001
										Apo B	2.08 ± 0.19	1.49 ± 0.13	−28.4	<0.01
Laurin et al., 1991 [17]	Canada	10 children	4F/6M	6–12	Crossover	None	4 weeks	Fully controlled diet with 35% of protein intake provided from cow’s milk	Fully controlled diet with 35% of protein intake provided from soy protein beverage	LDL-C	6.29 ± 0.37	6.33 ± 0.34	0.64	NS
										Total-C	7.89 ± 0.34	7.89 ± 0.34	0.0	NS
Wolfe et al., 1992 [18]	Canada	5 adults	3F/2M	48 ± 6	Crossover	Cholestyramine (18 g/d)	4–5 weeks	Fully controlled diet with 10% of energy as protein and 65% as carbohydrate	Fully controlled diet with 27% of energy as protein and 48% as carbohydrate	LDL-C	5.5 ± 0.5	5.3 ± 0.7	−3.6	NS
										Total-C	7.3 ± 0.7	7.1 ± 1.1	−2.7	NS
Fuentes et al., 2008 [19]	Spain	30 adults	30F/30M	42 ± 18	Crossover	Statins	16 weeks	Fully controlled diet with 280–300 mg/day of dietary cholesterol	Fully controlled diet with 150 mg/day of dietary cholesterol	LDL-C	4.01 ± 0.98	3.96 ± 1.09	−1.2	NS
										Total-C	5.92 ± 1.09	5.84 ± 1.24	−1.4	NS
Dietary counseling interventions														
Chisholm et al., 1994 [20]	New Zealand	19 adults	11F/8M	51 ± 10	Double-blind, crossover	Simvastatin	8 weeks	Exchange lists, recipes and nutritional counseling given every 15 or 30 days by an experienced dietician to adopt a high fat diet (38% energy from fat, 14% energy from saturated fat)	Exchange lists, recipes and nutritional counseling given every 15 or 30 days by an experienced dietician to adopt a low fat diet (27% energy from fat, 8% energy from saturated fat)	LDL-C	4.22 ± 0.93	3.95 ± 0.70	−6.4	<0.05
										Total-C	6.36 ± 0.98	5.96 ± 0.75	−6.3	<0.01
Broekhuizen et al., 2012 [21]	Netherlands	340 adults	193F/147M	18–70	Parallel	Yes, for n = 25 in the intervention group and n = 111 in the control group; type and dose not available.	52 weeks	Care as usual (n = 159)	Web-based advice, face-to-face counselling (n = 1) and telephone booster sessions (n = 1–5) with trained lifestyles coaches to adopt and maintain a healthier lifestyle regarding, saturated fat, fruits and vegetables intake (n = 181)	LDL-C	3.6 ± 1.2	3.5 ± 1.1	−2.8	NS
										Total-C	5.1 ± 1.2	5.2 ± 1.2	2.0	NS
Helk et al., 2019 [22]	Austria	34 children randomized (26 included in analyses)	18F/8M	4–14	Parallel	None	13 weeks	Counseling sessions led by an experienced dietician where practical advices on how to replace as many visible fat sources as possible with rapeseed oil were given	Counseling sessions led by an experienced dietician where recipes and practical advices on how to replace as many visible fat sources as possible with rapeseed oil and how to enrich diet in soy protein	LDL-C	4.65 ± 1.08	4.01 ± 0.78	−13.8	NS
										Total-C	6.58 ± 1.03	6.27 ± 0.96	−4.7	NS
										Apo B	1.23 ± 0.21	1.13 ± 0.19	−8.1	NS
Dietary supplement interventions														
Omega-3 supplementation														
Balestrieri et al., 1996 [23]	Italy	16 adults	7F/9M	45.2 ± 15.0	Double-blinded, crossover	Simvastatin (10–40 mg/d)	4 weeks	Olive oil capsules (6 g/day)	Fish oil ethyl ester capsules (6 g/day)	LDL-C	5.9 ± 1.3	5.9 ± 1.3	0.0	NS
										Total-C	7.8 ± 1.1	7.8 ± 1.3	0.0	NS
										Apo B	2.1 ± 0.4	2.0 ± 42	−4.8	NS
Chan et al., 2016 [24]	Australia	20 adults	10F/10M	18–70	Crossover	Statin or statin + ezetimibe	16 weeks	No placebo (usual diet)	ω-3 fatty acid ethyl ester supplements (4 g/day)	LDL-C	2.81 ± 0.29	2.54 ± 0.16	−9.6	NS
										Total-C	4.58 ± 0.27	4.20 ± 0.16	−8.30	NS
										Apo B	0.83 ± 0.06	0.76 ± 0.03	−8.4	0.04
Hande et al., 2019 [14]	Norway	38 adults	17F/17M	18–71	Double-blinded, crossover	Statins	12 weeks	Olive oil capsules (4 g/day)	ω-3 fatty acid supplements with 460 mg of eicosapentaenoic acid + 380 mg of docosahexaenoic acid (4 g/day)	LDL-C	3.2 ± 0.9	2.8 ± 0.9	−12.5	<0.01
										Total-C	5.0 ± 1.1	4.6 ± 0.8	−8.0	<0.05
Phytosterol supplementation														
Gylling et al., 1995 [25]	Finland	14 children	7F/7M	2–15	Double-blind, crossover	None	6 weeks	Rapeseed oil margarine without sitostanol ester	Rapeseed oil margarine enriched in sitostanol ester (3 g/day of sitostanol)	LDL-C	5.47 ± 0.30	4.65 ± 0.32	−15.0	<0.05
										Total-C	7.62 ± 0.32	6.81 ± 0.34	−10.6	<0.05
Neil et al., 2001 [26]	United Kingdom	62 adults	36F/26M	18–69	Double-blind, crossover	Statin	8 weeks	Placebo fat spread (25 g/day)	Plant sterol-enriched fat spread (25 g/day) providing 2.5 g/day of phytosterols (β-sitosterol: 50%; campesterol: 25%; stigmasterol: 20%; other sterols: 5%)	LDL-C	4.99 ± 1.02	4.65 ± 1.14	−6.8	0.001
										Total-C	7.20 ± 1.04	6.84 ± 1.12	−5.0	0.001
										Apo B	1.47 ± 0.29	1.46 ± 0.33	−0.7	NS
Amundsen et al., 2002 [27]	Finland	41 children	NA	7–12	Double-blind, crossover	None	8 weeks	Placebo fat spread (20 g/day)	Plant sterol-enriched fat spread (20 g/day) providing 1.60 ± 0.13 g/day of sterol esters (sitosterol: 50%)	LDL-C	5.88 ± 1.79	5.25 ± 1.55	−10.7	0.003
										Total-C	7.48 ± 1.70	6.87 ± 1.45	−8.2	0.007
										Apo B	1.48 ± 0.39	1.32 ± 0.35	−10.8	0.02
de Jonhg et al., 2003 [28]	Netherlands	41 children	21F/20M	9.2 ± 1.6	Double-blind, crossover	None	4 weeks	Placebo margarine (15 g/day)	Plant sterol spread (15 g/day) providing 2.3 g of sterols (sitosterol: 46.9%; campesterol: 27.3%; stigmasterol: 16.3%; other sterols: 9.5%)	LDL-C	5.40 ± 1.37	4.58 ± 1.13	−15.2	<0.001
										Total-C	7.06 ± 1.35	6.27 ± 1.12	−1.2	<0.001
O’Neill et al., 2004 [29]	United Kingdom	69 adults	39F/30M	53 ± 1.5	Double-blind, parallel (4 arms)	Statins or statins + bile acid sequestrants in n = 65/69	8 weeks	Placebo margarine (20 g/day)	1. Plant sterol-enriched fat spread (20 g/day) providing 1.6 g/day of free sterols + 1 placebo cereal bar (25 g/day)	LDL-C	3.81 ± 0.15	3.63 ± 0.15	−4.7	0.003
										Total-C	5.80 ± 0.17	5.50 ± 0.16	−5.2	<0.001
									2. Plant stanol-enriched fat spread (20 g/day) providing 1.6 g/day of free stanols + 1 placebo cereal bar (25 g/day)	LDL-C	3.83 ± 0.16	3.54 ± 0.14	−7.6	0.03
										Total-C	5.80 ± 0.19	5.50 ± 0.18	−5.2	0.006
									3. Plant stanol-enriched fat spread (20 g/day) providing 1.6 g/day of free stanols + 1 cereal bar (25 g/day) providing 1.0 g/day of stanol ester	LDL-C	3.77 ± 0.18	3.30 ± 0.14	−12.5	<0.001
										Total-C	6.1 ± 0.20	5.3 ± 0.15	−13.1	<0.001
Jakulj et al., 2006 [30]	Netherlands	41 children	19F/22M	7–12	Double-blind, crossover	None	4 weeks	Low-fat yogurt (500 mL/day)	Low-fat yogurt (500 mL/day) providing 2 g/day of plant stanols	LDL-C	5.24 ± 1.45	4.77 ± 1.32	−9.0	<0.001
										Total-C	7.00 ± 1.49	6.47 ± 1.35	−7.6	<0.001
Fuentes et al., 2008 [2,19]	Spain	30 adults	30F/30M	42 ± 18	Crossover	Statins	16 weeks	Fully controlled diet with 280–300 mg/day of dietary cholesterol and 0.5 g/day of plant sterols	Fully controlled diet with 280–300 mg/day of cholesterol and 2.5 g/day of plant sterols	LDL-C	4.01 ± 0.98	3.70 ± 0.98	−7.7	NS
										Total-C	5.92 ± 1.09	5.61 ± 1.01	−5.2	NS
										Apo B	1.13 ± 0.20	1.07 ± 0.20	−5.3	NS
Other types of supplementation														
Wirth et al., 1982 [31]	Germany	12 adults	5F/7M	30–60	Crossover	Bezafibrate (600 mg/d)	8 weeks	Usual diet	Granulated guar gum (15.6 g/day)	LDL-C	7.06 ± 1.81	6.08 ± 1.91	−13.9	< 0.01
										Total-C	9.08 ± 1.99	8.51 ± 1.78	−6.3	< 0.01
										Apo B	2.05 ± 0.21	1.55 ± 0.17	−24.4	NS
Greyling et al., 2006 [32]	South Africa	16 adults	14F/2M	47 (41.6–52.4)	Double-blind, crossover	None	12 weeks	Placebo capsules	Policosanol supplements (20 mg/day)	LDL-C	5.25 (95% CI: 4.26 –6.23)	5.47 (95% CI:4.56–6.38)	4.2	NS
										Total-C	7.07 (95% CI 6.–8.03)	7.21 (95% CI: 6.26–8.15)	2.0	NS

^1^ Values are expressed in mmol/L for LDL-cholesterol (LDL-C) and Total-Cholesterol (Total-C), and in g/L for Apolipoprotein B (Apo B).

**Table 2 nutrients-13-00235-t002:** Assessment of the risk of bias of included studies.

Study, Year	Randomization Process	Deviations from Intended Interventions	Missing Outcome Data	Measurement of the Outcome	Selection of the Reported Result	Overall Bias
Food-based interventions						
Gustafsson et al., 1983 [15]	Some concerns	Some concerns	Low risk	Low risk	Low risk	Some concerns
Friday et al., 1991 [16]	Some concerns	Some concerns	Low risk	Low risk	Low risk	Some concerns
Laurin et al., 1991 [17]	Some concerns	Some concerns	High risk	Low risk	Low risk	High risk
Wolfe et al., 1992 [18]	Some concerns	Low risk	Low risk	Low risk	Low risk	Some concerns
Fuentes et al., 2008 [19]	Some concerns	Some concerns	Low risk	Low risk	Low risk	Some concerns
Dietary counseling interventions						
Chisholm et al., 1994 [20]	Some concerns	High risk	Low risk	Low risk	Low risk	High risk
Broekhuizen et al., 2012 [21]	Low risk	High risk	Some concerns	Low risk	Low risk	High risk
Helk et al., 2019 [22]	Some concerns	Some concerns	High risk	Low risk	Low risk	High risk
Dietary supplement interventions						
Omega-3 supplementation						
Balestrieri et al., 1996 [23]	Some concerns	Low risk	Some concerns	Low risk	Low risk	Some concerns
Chan et al., 2016 [24]	Some concerns	High risk	Low risk	Low risk	Low risk	High risk
Hande et al., 2019 [14]	Low risk	Low risk	Low risk	Low risk	Low risk	Low risk
Phytosterol supplementation						
Gylling et al., 1995 [25]	Low risk	Low risk	Low risk	Low risk	Low risk	Low risk
Neil et al., 2001 [26]	Low risk	Low risk	Low risk	Low risk	Low risk	Low risk
Amundsen et al., 2002 [27]	Low risk	Low risk	Low risk	Low risk	Low risk	Low risk
de Jongh et al., 2003 [28]	Low risk	Low risk	Low risk	Low risk	Low risk	Low risk
O’Neill et al., 2004 [29]	Low risk	High risk	Low risk	High risk	Low risk	High risk
Jakulj et al., 2006 [30]	Low risk	Low risk	Low risk	Low risk	Low risk	Low risk
Fuentes et al., 2008 [19]	Some concerns	Some concerns	Low risk	Low risk	Low risk	Some concerns
Other types of dietary supplementation						
Wirth et al., 1982 [31]	Some concerns	High risk	Low risk	Low risk	Low risk	High risk
Greyling et al., 2006 [32]	Low risk	Low risk	Low risk	Low risk	Low risk	Low risk

## Data Availability

Data is contained within the article and the Appendix A.

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
