# Peer review of "Impact of Diet on Plasma Lipids in Individuals with Heterozygous Familial Hypercholesterolemia: A Systematic Review of Randomized Controlled Nutritional Studies"

_nutrients, 2021, doi:10.3390/nu13010235_

Round 1

Reviewer 1 Report

The aim of this study was to systematically review the impact on plasma lipids of previously published randomized controlled nutritional studies (RCTs) conducted in individuals with heterozygous familial hypercholesterolemia (HeFH), in order to assess the impact of food-based interventions, dietary counseling interventions, and dietary supplement RCTs on LDL-C and other plasma lipids in children or adults with HeFH in relation to their risk of bias. A total of 19 articles were thus included in the review: a significant reduction in LDL-C was observed in 10 of those, including 8 dietary supplement interventions, one food-based intervention and one dietary counseling intervention. As much as 13 studies were judged to have some methodological biases that substantially lowers confidence in the results. Overall, dietary interventions at low risk of bias were more likely to report significant reductions in LDL-C compared with those at risk of bias. in conclusion, this review suggests that the lack of conclusive data on the effectiveness of diet manipulation in decreasing LDL-C levels in HeFH is likely due to biases in conducted studies rather than a true lack of effects.

The paper is very interesting and well written

Only a minor point should be underlined: the numerosity of studies included in each subgroup reported in the abstract and in the text is different from that reported in table 1 (the food-based interventions are 5, and not 4; the phytosterols trials are 7, and not 6; line 130, "there were four food-based interventions (16-19)" did not include ref. 20; and similarly in other parts of the text. Please clarify

Author Response

Response to Reviewers – Manuscript Nutrients-1058507

We would like to thank the editorial team of Nutrients as well as the reviewers for their constructive comments on our manuscript “Impact of diet on plasma lipids in individuals with heterozygous familial hypercholesterolemia: a systematic review of randomized controlled nutritional studies”. We have been able to address all of the reviewer’s concerns and make the appropriate changes in the revised manuscript. Please find below our point-by-point response to the comments.

Reviewer 1:

The aim of this study was to systematically review the impact on plasma lipids of previously published randomized controlled nutritional studies (RCTs) conducted in individuals with heterozygous familial hypercholesterolemia (HeFH), in order to assess the impact of food-based interventions, dietary counseling interventions, and dietary supplement RCTs on LDL-C and other plasma lipids in children or adults with HeFH in relation to their risk of bias. A total of 19 articles were thus included in the review: a significant reduction in LDL-C was observed in 10 of those, including 8 dietary supplement interventions, one food-based intervention and one dietary counseling intervention. As much as 13 studies were judged to have some methodological biases that substantially lowers confidence in the results. Overall, dietary interventions at low risk of bias were more likely to report significant reductions in LDL-C compared with those at risk of bias. In conclusion, this review suggests that the lack of conclusive data on the effectiveness of diet manipulation in decreasing LDL-C levels in HeFH is likely due to biases in conducted studies rather than a true lack of effects.

The paper is very interesting and well written

Only a minor point should be underlined: the numerosity of studies included in each subgroup reported in the abstract and in the text is different from that reported in table 1 (the food-based interventions are 5, and not 4; the phytosterols trials are 7, and not 6; line 130, "there were four food-based interventions (16-19)" did not include ref. 20; and similarly in other parts of the text. Please clarify

Response: We would like to thank reviewer 1 for the positive appreciation of our study. We clarified the number of studies included in each subgroup throughout the text. One element that originally caused the confusion related to the number of included studies is that the study by Fuentes et al. (reference #20) was included in both the food-based interventions and the dietary supplement interventions because of its factorial design. We reformulated several sentences to make this element clearer.

 Please see changes we made in the abstract (page 1, lines 21+) and in the results section (section 3.1 Study selection, p. 3, lines 118+; section 3.2 Characteristics of the included studies, p. 4, lines 128+).

Reviewer 2 Report

The authors performed a systematic review of the literature on randomized controlled dietary interventions in heterozygous FH patients. They showed that studies at low risk of bias were more likely to report significant reduction in LDL-C levels compared with those at some concern or high risk of bias”. The studies at low risk of bias were often studies supplementing one nutrient in one product (e.g. plant sterol in a product replacing margarine). Dietary counseling interventions and food-based interventions tended to have a high risk of bias.

While the 2018 Canadian Cardiovascular Society statement on FH states that “conclusive data regarding the effectiveness of dietary interventions in FH are unavailable”, the authors conclude that their "systematic review first revealed that the lack of effectiveness of dietary interventions in modulating LDL-C levels in HeFH is likely due to biases in study designs". Furthermore, (some) studies with low risk of bias did show effectiveness. This effectiveness was in line with general knowledge on cardiovascular conditions.

The question is how to build policy for FH diet advice on the limited evidence. Apart from the specific evidence for FH, also general knowledge on cardiovascular morbidity needs to be considered.

The authors are involved in initiatives to study the Alternative Healthy Eating Index (AHEI) in FH, but how can observational evidence avoid bias? Or, as the authors state correctly´ the observational design precludes from inferring causality”.

The authors conclude that “There is thus a true need for high quality trials on the dietary impact on cardiovascular health in HeFH to unsettle this misconception about diet in this high-risk population.”

I agree to a large extent, however, I would suggest to separately discuss (1) Dietary interventions using supplements in margarine and (2) complex dietary interventions to simulate healthy diet. Furthermore, how much health gain can be achieved by medications, margarine vs. complex dietary interventions? If the additional risk in FH patients is already returned to normal with (early) statins, how much gain can be achieved (in terms of population attributable fraction or other epidemiological measures)?

Furthermore, what evidence from cardiovascular disorders in general can be used for FH dietary advice?

In Figure 1 typo “fßor”-> “for”

Author Response

Response to Reviewers – Manuscript Nutrients-1058507

We would like to thank the editorial team of Nutrients as well as the reviewers for their constructive comments on our manuscript “Impact of diet on plasma lipids in individuals with heterozygous familial hypercholesterolemia: a systematic review of randomized controlled nutritional studies”. We have been able to address all of the reviewer’s concerns and make the appropriate changes in the revised manuscript. Please find below our point-by-point response to the comments.

Reviewer 2:

The authors performed a systematic review of the literature on randomized controlled dietary interventions in heterozygous FH patients. They showed that studies at low risk of bias were more likely to report significant reduction in LDL-C levels compared with those at some concern or high risk of bias”. The studies at low risk of bias were often studies supplementing one nutrient in one product (e.g. plant sterol in a product replacing margarine). Dietary counseling interventions and food-based interventions tended to have a high risk of bias.

While the 2018 Canadian Cardiovascular Society statement on FH states that “conclusive data regarding the effectiveness of dietary interventions in FH are unavailable”, the authors conclude that their "systematic review first revealed that the lack of effectiveness of dietary interventions in modulating LDL-C levels in HeFH is likely due to biases in study designs". Furthermore, (some) studies with low risk of bias did show effectiveness. This effectiveness was in line with general knowledge on cardiovascular conditions.

The question is how to build policy for FH diet advice on the limited evidence. Apart from the specific evidence for FH, also general knowledge on cardiovascular morbidity needs to be considered.

The authors are involved in initiatives to study the Alternative Healthy Eating Index (AHEI) in FH, but how can observational evidence avoid bias? Or, as the authors state correctly´ the observational design precludes from inferring causality”.

The authors conclude that “There is thus a true need for high quality trials on the dietary impact on cardiovascular health in HeFH to unsettle this misconception about diet in this high-risk population.”

I agree to a large extent, however, I would suggest to separately discuss (1) Dietary interventions using supplements in margarine and (2) complex dietary interventions to simulate healthy diet. Furthermore, how much health gain can be achieved by medications, margarine vs. complex dietary interventions? If the additional risk in FH patients is already returned to normal with (early) statins, how much gain can be achieved (in terms of population attributable fraction or other epidemiological measures)?

Furthermore, what evidence from cardiovascular disorders in general can be used for FH dietary advice?

In Figure 1 typo “fßor”-> “for”

Response: We would like to thank reviewer 1 for the positive appreciation of our study and the important discussion points raised.

First, with regard to the question on how to build policy for FH diet advice based on the limited available evidence, we agree with the reviewer that we need to consider both evidence from individuals with FH as well as data on the relationship between diet and cardiovascular health in the general population. This is what we highlighted in the second paragraph of the discussion: “Seminal studies unequivocally demonstrated that diet is a major determinant of CHD risk across different populations including individuals with a high genetic susceptibility to CHD [34-36]. Moreover, multiple shorter-term RCTs demonstrated that diet also has an important impact on plasma lipids [37-39].” We also considered this element later in the discussion: “Notably, our work also highlighted that, in adequate study settings, diet appears to impact LDL-C and other plasma lipids similarly among individuals with HeFH than among non-FH individuals”

Still, evidence on the relationship between diet and cardiovascular health obtained from individuals with FH are much likely to be the cornerstone of dietary policies for FH relative to data from the general non-FH population. In this context, and as underscored in the discussion of our paper, high quality data on this subject are clearly needed. Our systematic review shed light on the fact that currently available high-quality data on the impact of diet on plasma lipids in FH support the importance of diet in FH management, but these remain sparse. On the other hand, as exemplified from the quote from the 2018 statement of the Canadian Cardiovascular Society, diet has traditionally been considered as an ineffective therapy in FH. In our opinion, building policy for FH diet advice first requires that scientific and medical organizations acknowledge the importance of healthy dietary habits in FH and its impact on cardiovascular health of these individuals. In that regard, we believe that our systematic review is an important contribution.

Next, the reviewer questions on how we can leverage observational data to build dietary recommendations for FH, even though we cannot infer causality from such evidence. This is, again, a very important question as nutritional epidemiology is often criticised for its reliance on observational studies to address etiologic questions. Randomized trials with hard endpoints occupy the highest position in evidence hierarchy, but they are not the most appropriate or feasible study design to answer nutritional epidemiologic questions regarding long-term effects of specific foods or nutrients. In the absence of evidence from large RCTs on hard endpoints, prospective cohort studies remain the strongest observational study design in terms of minimizing bias and inferring causality (Satija A et al. Adv Nutr 2015). Such studies are less affected by several biases, such as reverse causation, recall bias, and selection bias, which commonly plague cross-sectional studies, as per our investigation on the correlates of coronary artery calcification in FH where we documented for the first time an inverse relationship between diet quality and coronary calcification. Indeed, prospective cohort studies minimize the possibility of reverse causation because participants are followed in time and because prospective cohort studies begin with a disease-free population at baseline that is followed up to ascertain incident cases that develop over time. The reviewer is invited to read the paper by Satija et al. entitled “Understanding Nutritional Epidemiology and Its Role in Policy” published in Advances in Nutrition in 2015 for a thorough discussion on this subject. With regard to how we can leverage observational data to build dietary recommendations for FH, we strongly believe that prospective investigations on the relationship between diet and cardiovascular health in FH are required and will provide impactful evidence demonstrating the importance of healthy dietary habits in this high risk condition. These evidence would complement data from shorter-term RCTs that assessed the impact of diet plasma surrogate markers of cardiovascular health, like the studies we reviewed in the current work

Next, the reviewer questions on how much health gain can be achieved by medications, margarine vs. complex dietary interventions in FH, and what evidence from cardiovascular disorders from the general population can be used for FH dietary advice. The former is challenging to quantify per se using metrics such as PAF, because of the lack of data on the cardioprotective potential of diet in FH. Still, we strongly believe that healthy dietary habits are as much important as medication in FH management. Also, we stress that medication and diet should be used in a complementary fashion in order to maximize the effects of both. In that regard, our work highlighted that, in adequate study settings, diet impacts LDL-C and other plasma lipids similarly among individuals with HeFH than among non-FH individuals. Indeed, six of the seven interventions on phytosterol (sterols and/or stanols) supplementation we reviewed reported significant reductions in LDL-C among individuals with HeFH, ranging from 5% to 15%. These reductions are consistent with mean effects on LDL-C reported in meta-analyses on phytosterol supplementation interventions conducted among non-FH individuals. Similarly, substituting unsaturated fatty acids for saturated fatty acids in the food-based intervention conducted by Friday et al. decreased LDL-C levels by about 30%. The observed cholesterol-lowering effects associated with dietary fat quality manipulation was also consistent with data from studies conducted among non-FH individuals. Lastly, with regard to omega-3 supplementation, no significant effect on LDL-C levels was reported in two of the three retrieved studies, but reductions in TG levels of >15% were reported in all of them. Again, these findings are consistent with data among non-FH individuals as omega-3 fatty acids, relative to other unsaturated fatty acids, are recognized to lower TGs levels, not LDL-C. These data are thus strongly supportive of an important for of diet in FH management. Still, how these effects translate into CVD risk remain to be assessed.

Lastly, we agree with the reviewer that . On the other hand, a large inter-individual variability in the response to statins is reported, with only 50% of HeFH patients achieving LDL-C targets. This exposes about half of patients to a high residual CVD risk. Besides, diet, relative to medication, impacts not only plasma lipids, but also inflammation, glucose homeostasis and blood pressure, all important CHD risk factors in HeFH. Overall, we stress that medication and diet should be used in a complementary fashion in FH management in order to maximize the efficacy of medication efficacy and attenuate CVD risk in this high-risk population.

In line with the comments raised by the reviewer, we modified parts of the discussion to better contextualize the place of diet in FH management. Please see changes made on:

  • Page 15, first paragraph of the discussion: “In this context, diet should be recognized as a factor significantly affecting plasma lipids in HeFH.”
  • Page 16: “[51,52]. On the other hand, a large inter-individual variability in the response to statins is reported, with only 50% of HeFH patients achieving LDL-C targets [53]. This exposes the other half of patients to a high residual CVD risk [54,55].”
  • We corrected the typo in Figure 1.